# Examining the alignment between subjective effort and objective force production

**Katja Rewitz**[1,2]\*, **Sebastian Schindler**[3], **Wanja Wolff**[1,2]

**1** Dynamics of Human Performance Regulation Laboratory, Institue of Human Movement Science, University of Hamburg, Hamburg, Germany, **2** Department of Sports Science, University of Konstanz, Konstanz, Germany, **3** Institute of Medical Psychology and Systems Neuroscience, University of Münster, Münster, Germany

\* katja.rewitz@uni-hamburg.de

**Data Availability Statement:** All data and materials can be found at OSF | alignment subjective effort_objective force production. See here: https://osf.io/9dyxn/,

## Abstract

Ratings of Perceived Exertion (RPE) are frequently used to prescribe exercise intensity. A central assumption of using RPE scales is that the subjective perception of effort maps onto objective performance in a consistent way. However, the degree and shape of how RPE aligns with objective performance is not fully understood. Here, we investigate the degree and shape of alignment, as well as how time (i.e., how frequently an effort needs to be performed) and mental effort (i.e., if one has to invest mental effort and physical effort) correspond with the alignment. In a randomized within-subjects experiment, we used a grip-to-scale method that asked participants (N = 43) to repeatedly squeeze a handgrip dynamometer with four to-be-produced RPE target levels relative to their subjective maximum strength (representing 20%, 40%, 60%, or 80%). We found that the RPE-force alignment was not the same across RPE-levels: Whereas subjective differences from 20–40% and 40–60% were met by comparable differences in produced force, a substantially larger difference was observed for the 60–80% interval. Interestingly, exploratory post-hoc analyses revealed that this was mirrored by an increase in variance at the higher effort levels. In addition, at constant RPE-levels, participants produced less force over time, and this effect was more pronounced at lower RPE target levels. Lastly, anticipating mental effort after the physical effort slightly altered the alignment as a function of the to-be-produced RPE-level and experimental duration. Taken together, our results indicate that the mapping of perceived effort on objective performance is intricate, and several factors affect the degree and shape of how RPE and performance align. Understanding the dynamic adjustment of RPE-performance alignment across different RPE levels is particularly relevant for contexts that use RPE as a tool for training load prescription.

## Introduction

Physical exercise is associated with a reduction of health-related risk factors, higher quality of life, and reduced mortality [1–3]. Mental exercise is a frequently used lay term describing the engagement in mentally stimulating activities and is suggested to contribute to developing a "cognitive reserve". This reserve is posited to be a protective factor against cognitive decline

**Funding:** The author(s) received no specific funding for this work.

**Competing interests:** The authors have declared that no competing interests exist.

and clinical symptoms in the early onset of neurological diseases such as Alzheimer's [4, 5]. However, going for a run (physical exercise) or solving a complex mathematical problem (mental exercise) opposes most peoples' default/habitual response, which might be to stay on the couch or scroll through social media. Doing something at odds with one's default response requires cognitive control [6, 7]. Cognitive control, also referred to as self-control, is defined as "the set of mechanisms required to pursue a goal, especially when distraction and/or strong competing responses must be overcome" [8, p. 217]. Applying control is perceived as effortful [7], and perceived effort is understood to index the costs of applying control [9]. In the present study, we adapt the definition of perceived effort as "the instantaneous experience of utilizing energy to perform an action" [10, p. 8]. In line with this definition we understand the perception of effort to be based on the integration of various contributing factors rather than being solely contingent on objective behavioral output (e.g., force production) [10].

Motor control and cognitive control theories suggest that the execution of a control action reflects a reward-based choice that is geared towards minimizing control costs [8, 11]. To illustrate, according to one influential framework—the Expected Value of Control Theory (EVC theory)—the expected value of control is calculated for all available behavioral options by discounting the expected positive and negative payoffs that are associated with task engagement and the intrinsic cost of control. These intrinsic costs scale with task demands and task duration [8]. Thus, harder and longer tasks are expected to be more costly than short and easy ones [12]. During task performance, environmental information (e.g., feedback, opportunity for rewards) and internal sensations (e.g., pain, motivational loss, boredom) are assumed to be monitored, updated, and integrated to initiate adjustments in effort allocation [8, 13, 14]. Consequently, if changes in perceived control costs alter this cost-benefit analysis, performance decrements or even task termination are expected to ensue [8].

Empirical evidence demonstrates that the regulation of mental and physical effort can be effectively described by theories relying on the concept of a cost-benefit analysis [15, 16]. However, recent theoretical and empirical advancements notwithstanding, the relationship between perceived effort as an index of control costs and the translation of this perception into objective behavioral output is not fully understood. Simply put, we do not know how perceived effort levels and objective output align and what factors affect this alignment. In this study, we will investigate these questions through the lens of reward-based choice theories of control.

## How are perceived effort and behavioral output aligned?

Does an increase in perceived effort during a handgrip task lead to an equivalent increase in produced force? Some research shows that equivalent differences in perception of effort translate to equivalent differences in heart rate [17, 18]. However, from a conceptual point of view, it is important to note that equivalent differences in subjective exertion do not necessarily need to translate into equivalent differences in objective performance outcomes or physiological indicators of physical strain: Psychometrically, RPE scales are designed as category-ratio scales that measure differences in the perception of effort on an interval scale. However, this perception likely integrates various physical and psychological perceptual cues, and is not solely dictated by a single physiological parameter [17, 19]. In turn, the alignment between RPE and a singular objective measure of performance might not be linear (e.g., because different additional RPE-determining cues might be integrated at different levels of perceived effort). Indeed, studies that have used performance measures like handgrip strength indicate that the alignment follows an exponential function [19–21]. The bulk of this research uses a combination of estimation (i.e., participants are asked to rate their RPE regarding their physical performance) and production methods. Only a few studies exclusively focus on the

production method [22]. However, RPEs are not only used to ask participants for their perceived effort after stimulus presentation but are also critical for prescribing and monitoring exercise intensity [23, 24] for example in clinical rehabilitation and sports contexts (e.g., cycling, running and resistance training) [25–27]. This involves a production method where people are instructed to generate force at different subjective effort intensities, with the produced force being the dependent variable of interest. Instructions for intensity prescriptions are typically relative to a previously measured individual's maximal capacity, such as "Press 20% of your maximal possible force" [28–31]. This practice implies an assumption of a linear alignment between perceived effort and to-be-produced force, with participants using their perceived maximal force as a benchmark and scaling other perceived intensities relative to that (in turn, coaches would likely expect that their athletes train at the specified levels of physical load). Studies that used such production tasks reported heterogeneous results with respect to the translation of subjective effort into objective performance [28, 29, 31]. To illustrate, one study observed that up to 80% of participants' maximum voluntary contraction (MVC), subjective effort, and objective performance were aligned in a linear fashion, indicating equivalence of differences. However, equivalence was not found at higher intensities [29]. Similarly, another study found differences in the alignment across effort levels as a function of the study population [31]. Because of this mixed evidence and relatively few studies using a production method, a deeper understanding of how increases in perceived effort map onto changes in produced force is crucial.

## What is the effect of time on the alignment of perceived effort and behavioral output?

When squeezing a handgrip dynamometer with 50% of subjective maximum force a 100 times, it is conceivable that the produced force will decline over time because one is getting fatigued or bored by the task. Increasing cost perceptions are associated with reallocating resources and potentially decreasing the invested effort [8]. For example, increases in locomotor fatigue [32, 33] or the rising opportunity costs [34] might lower the objective force a person produces even though the person is producing effort levels that feel the same. Surprisingly, time is a largely overlooked variable when examining the alignment between perceived effort and actual force production. Typically, participants are required to produce the target levels somewhere between one to five times [28, 35]. However, since people usually need to complete a task more than once or twice, it is crucial to better understand the temporal dynamics of how subjective effort and objective performance align.

## Does prospective mental effort affect the alignment of perceived effort and behavioral output?

A large body of research has investigated the interplay between physical and mental effort regulation [12, e.g., 36–38]. On a conceptual level, this research points towards similarities in the underlying cost-benefit analyses [15, 39] that govern the decision of whether to employ effort [8, 40] or instead withhold and conserve it [41], as well as to similar neurocomputational underpinnings [9, 42]. Simply put, irrespective of the specific effort-domain (physical or mental), people seem to minimize its exertion and make exertion levels contingent on the expected rewards. This begs the question if/how the costs of either effort domain carry-over to the other domain. For example, researchers have investigated the effects of prior mental effort on subsequent physical performance [e.g., 36, 37] and the effects of concurrently performing a mental effort task on physical performance [e.g., 38]. These studies tend to show that prior mental effort can undermine subsequent physical effort and that concucrrent exertion of effort in the mental and physical domain can impair performance. However, to the best of our knowledge,

research has not yet investigated the potential effect of prospective mental effort on the regulation of physical effort. In light of the similarities in how physical and mental effort are governed, it is plausible that prospective efforts in one domain could affect regulation in another domain. For example, a person's effort during a run might be affected by the expectation of having to study for a demanding exam afterwards. As people strive to conserve resources, it is plausible that prospective efforts in one domain affect effort exertion in another domain.

### The present study

Here, we investigated the equivalence of alignment across effort levels (i.e., does a 20% difference in perceived effort translate to a 20% difference in produced force?) with a production task "grip-to-scale" method [43]. Participants repeatedly squeezed a handgrip dynamometer with four to-be-produced target levels relative to their pre-experiment maximal voluntary contraction (MVC). These target levels asked participants to produce efforts that subjectively felt like 20%, 40%, 60%, or 80% of their MVC.

First, we tested if equivalent differences in subjective effort map onto equivalent differences in produced force. It is plausible that the perceived effort costs (e.g., pain) do not change in equivalent steps across ascending RPE-levels. We, therefore, expected to replicate the non-equivalence in the alignment between subjective effort and produced force. Second, to investigate the temporal dynamics of how subjective effort and produced force align, participants squeezed the dynamometer 240 times (each force level was produced 60 times). Over time, factors such as locomotor fatigue, opportunity costs of time, or simply boredom were believed to accumulate and contribute to the cost perceptions. Therefore, while participants maintained a consistent subjective effort (for each to-be-produced level), we expected to observe a gradual reduction in produced force throughout the experiment. Third, we investigated whether prospective mental effort would affect the produced physical force. Therefore, we manipulated whether a mental task had to be performed after squeezing the dynamometer by announcing it prior to each handgrip trial. This allowed us to investigate the conserving and withholding of effort across task domains when the reward is held constant but the perceived costs change due to the manipulation of task difficulty. Because people are understood to optimize effort allocation to maximize the expected value associated with task engagement, we expected that physical resources are prospectively conserved as a function of subsequent mental demands.

## Methods

### Participants

A total of N = 44 adult university students (f = 18, m = 26) took part in the study (M = 23.96 ± 2.66 years old, Min = 18, Max = 32). Participants were recruited to participate in a study, where they receive a flat payment of 20€ for their participation. Only participants who spoke German as their first language and had no preexisting hand or arm injuries were included. No further inclusion or exclusion criteria were used. Prior to the experiment, informed consent was provided by all participants. The Institutional Review Board (Ethics Committee) of the University of Konstanz approved the experimental protocol and all used methods and adhered to the Declaration of Helsinki. All data and materials can be found at OSF | alignment subjective effort_objective force production.

### Design, measures, and procedure

The experiment was conducted as a randomized 4 (physical effort levels) × 3 (mental effort condition) within-subject design. All participants performed each of the 12 task combinations,

consisting of the four physical effort levels (2, 4, 6, 8) combined with the mental effort conditions (no-, low-, high-effort) 20 times.

**Measure of maximal voluntary contraction.**   To standardize the costs between participants, their MVC was measured. A Hand Grip Force Dynamometer (MLT004/ST Grip Force) combined with a digital PC interface (PL2516) and the data acquisition software LabChart (version v8.1.16) were used. The signal was sampled with a frequency of 10 Hz. The force was measured in Newtons. MVC was measured three times for approximately three seconds. Isometric hand grip MVC was defined as the peak force produced during three maximum contraction trials, whereas the measured force for the submaximal contractions, produced during the task, was averaged over two seconds. These differences in calculation were made on the assumption that, using a production method, participants need time to regulate their force to produce a certain target RPE-level, which cannot be captured by one peak value. Therefore, due to these differences in force calculations, the produced force is not expected to reflect the equivalent percent relative to MVC. Simply put, if participants are asked to produce 60% of MVC, participants are likely to produce less than 60%.

**Grip-to-scale measure.**   Participants were asked to produce four force levels in alignment with predetermined target levels of a modified Borg CR-10 scale [17]. The Borg CR-10 scale is a 10-point category-ratio-scale used to measure exertion and pain [44]. The upper anchor of 10 represents the maximal possible exertion. The target levels 2, 4, 6, and 8 were selected to cover nearly the whole scale range. Participants were instructed that these levels should represent 20%, 40%, 60%, and 80% of their maximum. The target levels were presented in a random order throughout the experiment. Participants were instructed that they should produce and maintain this force for the whole 4s period. During the whole experiment, participants performed 240 submaximal contractions.

**Mental tasks.**   Here, participants performed an N-back task [45], which consists of a quickly presented word stream. The participant is asked to respond by pressing the space bar when the same stimulus repeatedly appears, separated by the factor of N. To vary cognitive demands, two different task difficulties were used. The easy task (low-effort condition) consisted of a simple N-back task. Participants were instructed to respond if the same word appeared twice in a row. The difficult mental task (high-effort condition) was a 2N-back task. Here, a response was required when the same word was displayed, separated by another word. Participants performed 80 easy and 80 difficult mental tasks in the experiment. No mental task was performed in the remaining 80 trials (no-effort condition). For displaying the instructions and the mental tasks, MATLAB (version R2021a) was used [46].

**Setup.**   All contractions were performed, sitting in a neutral posture, legs on the ground, in front of a table on a desk chair. The subjects were instructed to hold the handgrip dynamometer vertically with their dominant hand, cable towards the bottom with the forearm outstretched at a 90°-angle. The instructions for the physical and mental tasks were displayed on a laptop in front of the participant. The spacebar of this laptop was used to indicate the reactions to the mental tasks. For responding, the participants should use their non-dominant hand.

**Procedure.**   Participants were instructed to sit in front of the prepared setup and pick up the handgrip dynamometer. They were told to avoid leaning towards their active arm while squeezing. At first, MVC was measured three times in a row, separated by a ten-second resting interval. Following this baseline measure, the exact procedure was explained to the participants. They were told to produce forces subjectively representing 20%, 40%, 60%, and 80% of their maximum, representing a subjective effort of 100% or a 10 on the RPE scale. To facilitate understanding, participants were presented with the modified CR-10 scale, which displayed the respective percentages (20%, 40%, 60%, 80%) next to the target levels (2, 4, 6, 8). All individual target levels were accompanied by verbal descriptors (not strenuous, slightly strenuous,

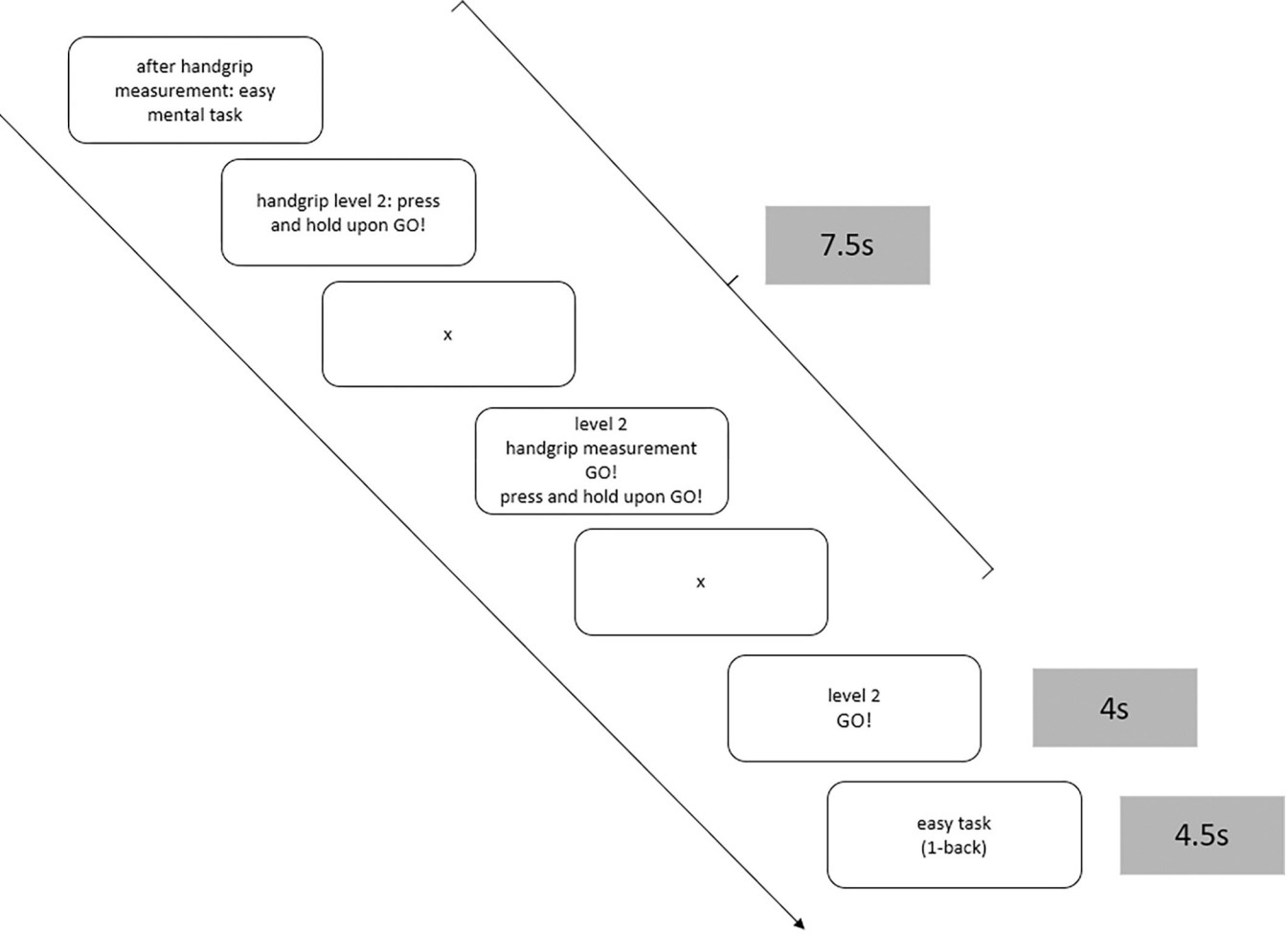

**Fig 1. Schematic representation of an experimental trial.** The combination of target level 2 and the easy mental task is illustrated. A trial lasts for 16 seconds, 7.5 seconds of which were used for instructions. The handgrip is measured for 4 seconds. During the subsequent 4.5 seconds, participants either perform the mental task (easy or difficult) or take a short break for the same duration (no mental task).

strenuous, very strenuous). They were instructed to squeeze for four seconds while the term "GO!" was displayed. For force productions participants should solely rely on their subjective feelings. No feedback was provided on either the MVC measures or the submaximal measurements. Prior to the detailed squeezing instruction, participants were informed whether there would be a subsequent mental task. This mental task was announced as either easy or difficult (Fig 1). In the control condition, no mental task was announced, which means that in this condition, participants had a short break after the handgrip measurement. The mental task condition lasts 4.5 seconds (Fig 1).

The experiment began after participants completed a practice N-back trial to familiarize themselves with the task. This trial could be repeated as often as necessary until the participant fully understood the task. There were no practice trials for the physical effort task. The full experiment consisted of 6 blocks of 40 trials, each with a duration of 16 seconds. Blocks were separated by a resting period of 150 seconds to avoid excessive fatigue.

After each block, performance feedback was given regarding the accuracy of responses across all mental tasks from the preceding experimental block. Feedback was provided as a self-monitoring device so participants could make sure that they were correctly performing

the different mental tasks throughout the 2-hour experiment. The feedback provided was not tied to specific trials. Hence, participants were not given any information on their performance regarding different task difficulties. Importantly, in order to make participants truly rely on their internal sensations of effort, no feedback or monitoring of their physical performance was provided. Further, after each block, subjects were asked to separately rate their perceived mental and physical exhaustion, each on a 9-point-rating scale ranging from 1 ("not at all") to 9 ("totally"). At the end of the 6 blocks, overall feedback was provided for the mental effort tasks. After the data collection, participants were debriefed, and compensated. The whole experiment lasted around 120 minutes.

## Statistical approach

MVC was operationalized as the peak force during the three MVC squeezes. The data of one participant was excluded due to technical difficulties in the handgrip recordings. Therefore, data analyses were performed with a sample of N = 43.

Since participants needed time to react to the instruction and also to stop, the first and last second (ten rows each) were excluded from the 4-second hand-grip measurements (40 rows), leaving the inner 2 seconds (20 rows) of the measurement for further analyses. When the data sets of the whole sample were combined, the mean force was averaged over the obtained 2 seconds, and the percentage of MVC ($MVC_{perc)}$) and the percentual deviation from the target level were calculated. To investigate whether the subjective differences of the RPE scale intervals transform into the same intervals in produced force, a one-way repeated measures ANOVA was performed with the averaged differences in produced force (%MVC) between target RPE levels as the dependent variable. Thus, we compared whether the relative difference in produced force for the three equal-sized subjective intervals of effort (i.e., 20–40%, 40–60%, and 60–80%), each reflecting a 20% increase in subjective effort, translate into equivalently sized intervals in produced force.

For further data analysis, a linear mixed model was used to account for the expected high inter- and intraindividual variability in force production and the dependency of the handgrip measurements. All analyses were performed using R and RStudio [47]. The model was computed with the lme4 package (version 1.1–29) [48]. Satterthwaite's approximation was used to estimate degrees of freedom for the fixed effects [49]. The model fitted fixed slopes for the independent variables "mental task", "RPE level" and "Nth event". The Nth event represents the 20 occurences for each task combination of physical effort levels (2, 4, 6, 8) and mental task condition (no-, low-, high-effort) and, therefore, serves as time variable. Because of high inter-individual differences in handgrip strength and time effects on force production, random slopes for the time variable and random intercepts for each participant were fitted (Model: $MVC_{perc}$ ~ mental task *RPE level * Nth event + (1 + Nth event | ID)). For each participant, we have 20 squeezes per condition, totaling 240 observations per subject. In post-hoc contrast comparisons, we, therefore, compare per subject each squeeze with the respective squeeze in the other comparison, thereby substantially increasing the number of computed comparisons. Bonferroni-corrected contrast analyses were performed by using the emmeans function of the same named package (version 1.7.5) [50].

## Results

The average MVC was 402.28N (SD = 122.15N), with a range of 213.98N – 651.63N. Relative to their MVC, participants produced M = 8.81% (SD = 6.66%) in effort target-level 2,
M = 15.07% (SD = 8.59%) in target-level 4, M = 20.69% (SD = 10.53%) in level 6 and
M = 33.14% (SD = 14.18%) in target-level 8. Participants reported overall increasing fatigue

over the task duration. Mental fatigue increased significantly from the first block (M = 2.79, SD = 1.71) to the sixth block (M = 4.98, SD = 1.99) (t(42) = 9.18, p < .001). Similarly, physical fatigue ratings showed a significant increase from after the first block (M = 3.53, SD = 2.00) to the end of the experiment (M = 5.19, SD = 2.26) (t(42) = 5.26, p < .001). However, across all blocks, participants reported a moderate fatigue level in the mental (M = 4.10, SD = 1.98) and physical (M = 4.47, SD = 2.08) domains, indicating that the procedure caused a moderate overall fatigue level. This indicates that participants were unlikely to be too exhausted to produce the required forces throughout the experiment. In turn, potential declines in produced force are unlikely to be purely due to accumulated physical fatigue, but might also reflect changes in participants cost-benefit analyses for exerting effort towards the task. Participants rated the perceived difficulty of the two N-back tasks on a 5-point Likert scale, ranging from 1 ("very difficult") to 5 ("not difficult at all"). Results of a paired sampled t-test showed a significantly higher perceived difficulty rating in the N-back task (t(42) = 12.071, p < .001). That is, participants rated the N-back task (M = 4.744, SD = 0.492) as less difficult than the 2N-back task (M = 3.721, SD = 0.701), indicating the manipulation of the mental effort conditions was as intended.

## Alignment of subjective effort level and objective force

To test the alignment of to-be-produced subjective effort level and objective power output, the linear mixed model revealed a significant main effect of physical effort level (F(1, 10224) = 3413.47, p<. 001, $\eta_p^2$ = 0.25). As expected, Bonferroni corrected post-hoc paired sample t-tests showed higher forces in level 4 compared to level 2 (t(2579) = 53.82, p < .001), in level 6 compared to level 4 (t(2579) = 39.63, p < .001) and in level 8 compared to level 6 (t(2579) = 62.73, p < .001), which indicates that the subjectively different efforts led to differences in produced force for each level of perceived effort. In physical effort level 2, force production differed M = 11.19% (SD = 6.66%), in level 4 M = 24.93% (SD = 8.59%), in level 6 M = 39.31% (SD = 10.53%) and in level 8 M = 46.86% (SD = 14.18%) from target level MVC. This underproduction relative to peak MVC is likely due to the fact that target level MVC was calculated as a proportion of the achieved peak force, whereas force production was averaged across 2 seconds. A significant difference was found when comparing the averaged deviations between the RPE-level deviates (F(1.36, 57.14) = 63.85, p < .001, $\eta_g^2$ = 0.34). The deviations represent the differences in the averaged produced force (in % MVC) between target level 20% compared to 40% (M = 6.26%, SD = 3.18%), 40% compared to 60% (M = 5.62%, SD = 3.74%) and 60% compared to 80% (M = 12.45%, SD = 5.58%). Bonferroni corrected post-hoc analyses revealed a significant difference in produced force relative to MVC between the target levels 20% and 40% compared to 60% and 80% (t(42) = -7.43, p < .001) and between the target levels of 40% and 60% in comparison with 60% and 80% (t(42) = -9.92, p < .001). The deviation between the 20% and 40% target levels did not differ from the deviation between the target levels of 40% and 60% (p = .39) (Fig 2). Error bars in Fig 2 indicate that the variances in produced forces are not the same for each effort level. Consequently, we conducted an unplanned exploratory post-hoc analysis to assess whether variances differed significantly between levels. Consistent with the descriptive trends in Fig 2, a repeated measures ANOVA and post hoc comparisons revealed that variance increased at higher effort levels F(1.32, 55.26) = 67.96, p < .001), with a significantly higher variance at target level 80% compared to the lower effort levels (p < .001). For the full analysis, please see OSF | alignment subjective effort_objective force production.

## Temporal force decline

To investigate the temporal dynamics, we examined whether force production changed over time when the to-be-produced subjective effort was constant. A significant time × physical

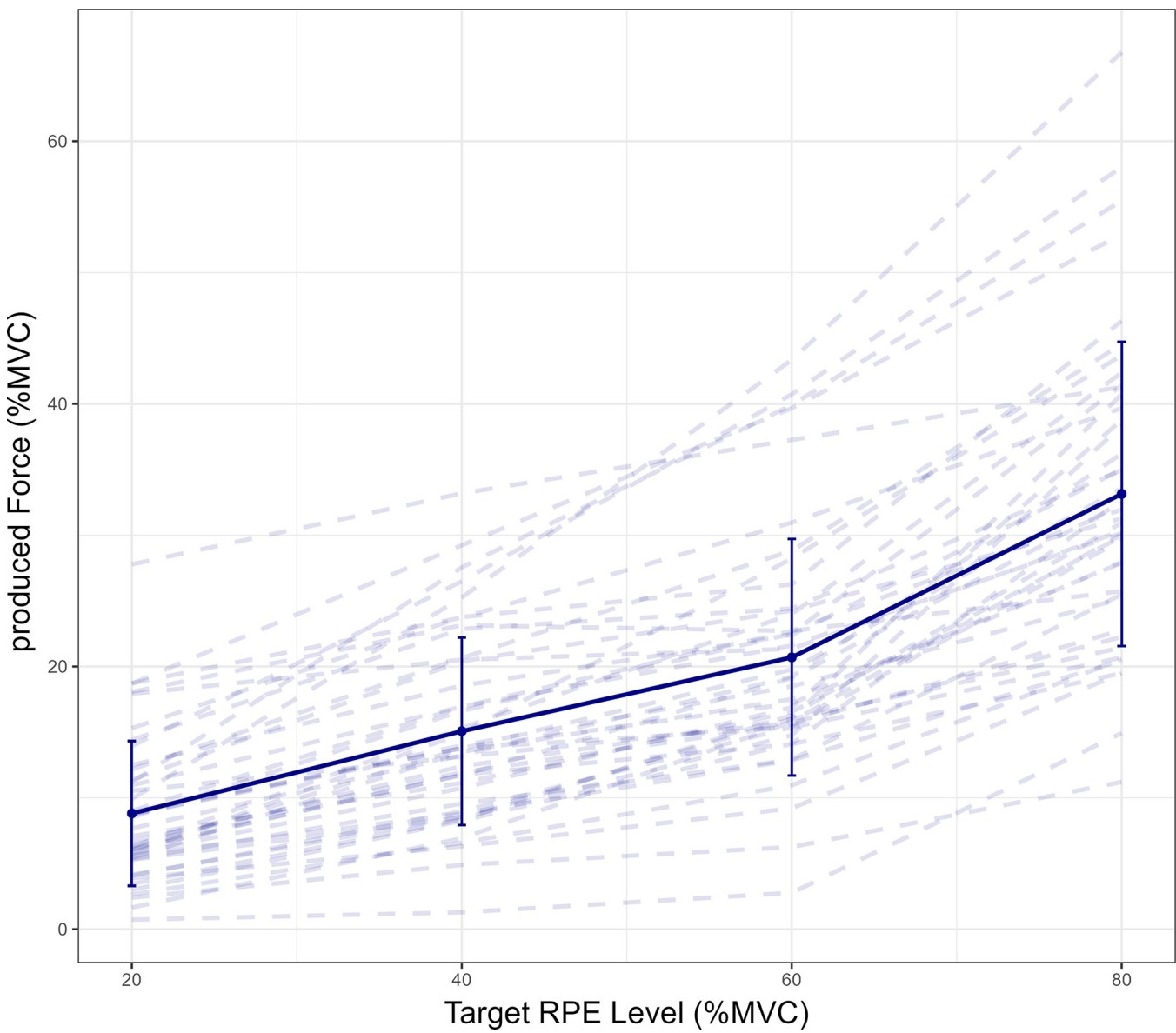

**Fig 2. Produced force relative to maximum voluntary contraction in each RPE target level.** The difference between each RPE-level represents a 20% increase in subjective effort. As the differences between the compared RPE-levels are equivalent, equivalence in translating subjective effort to objective performance would require a linear increase of produced force across RPE target-levels. However, the deviation between target levels 60% to 80% significantly differs from the other two deviates (20–40, 40–60), resulting in a nonlinear pattern. This indicates that equivalent differences in subjective efforts do not translate to equivalent differences in produced force across all effort levels. N = 43. Error bars represent the standard deviation.

effort level interaction effect ($F(1, 10224) = 10.68$, $p = .001$, $\eta_p^2 = 0.001$) indicated a change in force generation over time. Simple slope analyses revealed a significant force decrease in physical effort level 2 ($b = -0.21$, SE = 0.06, $p < .001$) and in level 4 ($b = -0.13$, SE = 0.06, $p = .03$). Slopes in level 6 and level 8 did not differ significantly from zero. This means that when participants were asked to produce lower levels of force (i.e., 20% and 40%), the actual force output declined over time, whereas no such decline was observed when higher levels of force were to be produced (i.e., 60% and 80%).

### Regulation of physical and mental effort

Anticipating a subsequent mental task did not lead to different physical performance. Testing for temporal dynamics in force generation when anticipating mental effort demands, the model revealed a significant but very small three-way-interaction effect of physical effort level, mental task condition, and the time variable ($F(2, 10224) = 4.91$, $p = .007$, $\eta_p^2 = 0.0009$) (Fig 3). Contrast analyses of the interaction effect revealed a significant three-way interaction in physical effort levels 4, 6, and 8 in the beginning and towards mid-experiment. To investigate differences between the mental effort conditions, simple slopes were compared. A priori, it was determined that simple slopes would be compared in the beginning (Nth event = 1), mid-experiment (Nth event = 10), and in the end (Nth event = 20). In the beginning, compared to the no-effort condition, analyses revealed significantly steeper slopes of the low mental effort condition in physical effort level 6 ($t(10224) = 4.47$, $p < .001$) and level 8 ($t(10224) = 4.93$, $p < .001$), as well as in the respective comparison of no mental effort and high mental effort in level 6 ($t(10224) = 2.35$, $p = 0.02$) and level 8 ($t(10224) = 2.14$, $p = 0.03$). The same pattern in the

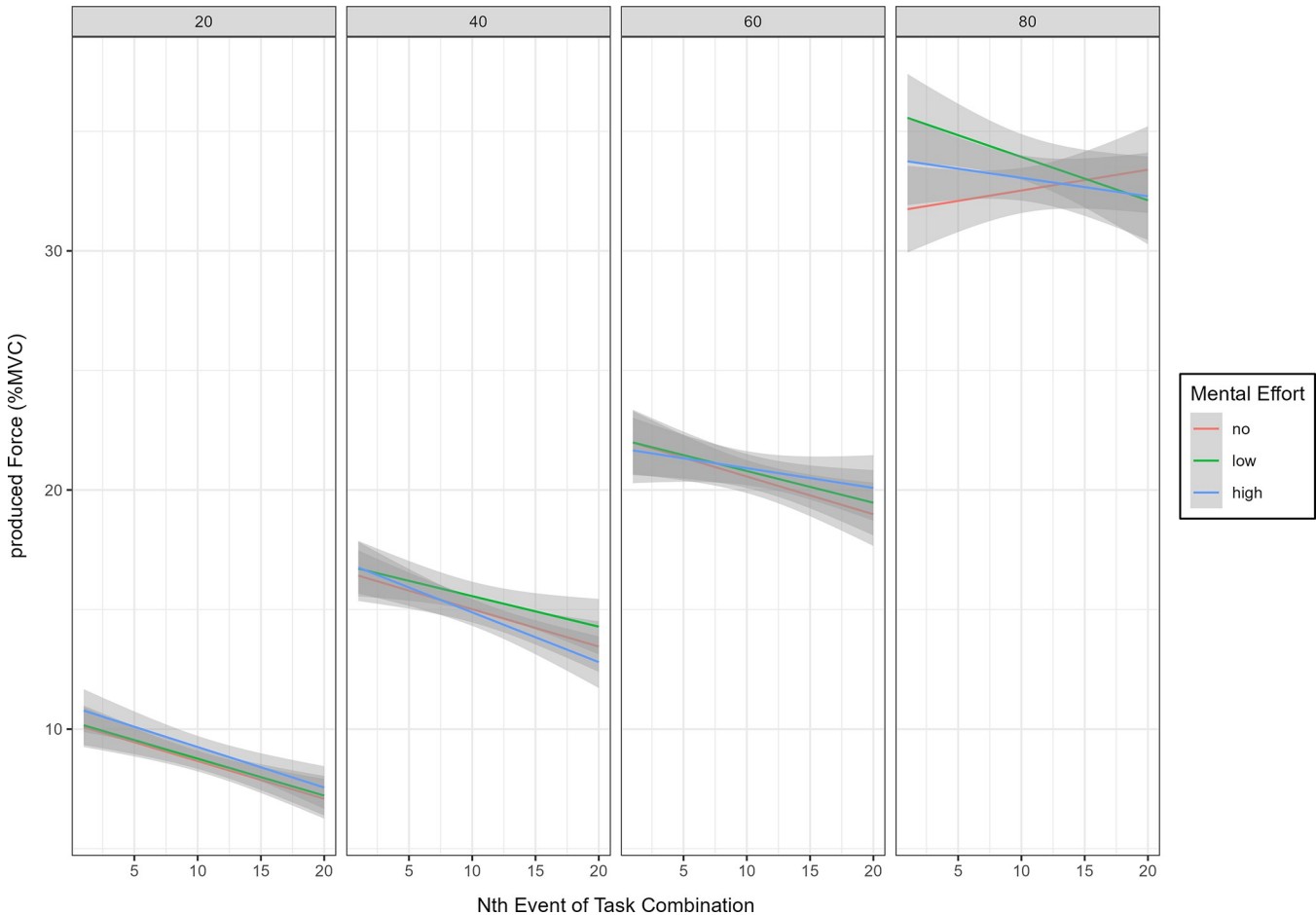

**Fig 3. Force generation over time, separated by physical effort level and mental effort condition.** N = 43. Scaled submaximal voluntary force production under to-be-produced effort target RPE-levels (%MVC). Shaded areas represent 95% confidence intervals. The generated force differed significantly between the physical effort levels. A significant three-way interaction was revealed in physical effort levels 40%, 60%, and 80%. Between conditions, comparisons revealed steeper slopes in the low and high mental effort conditions compared to the no mental effort conditions at the beginning of physical effort levels 60% and 80%. Additional between-condition comparisons revealed steeper slopes in the low mental effort condition compared to no mental effort towards mid-experiment in levels 40%, 60%, and 80%.

comparison of low and no mental effort was observed mid-experiment in level 6 (t(10224) = 4.04, p < .001), level 8 (t(10224) = 3.90, p < .001), but also in level 4 (t(10244) = 2.11, p = 0.03). Thus, in the beginning, and towards mid-experiment, participants squeezed less in some physical effort levels when no mental effort was anticipated compared to the expectation of upcoming mental effort.

## Discussion

Here, we investigated the alignment between subjective effort perception (RPE expressed in subjective % of one's maximum capacity) and force production. With the term alignment, we refer to the degree and shape of how one's feelings (i.e., "how hard am I squeezing this hand-grip dynamometer, relative to my maximum strength?") translate to objective performance (i.e., how much force is actually produced, relative to the person's maximum voluntary contraction?). Shape refers to whether the degree of alignment is consistent over different effort levels or a longer period. Here, we investigate alignment across different RPE-levels, time periods, and domains.

First, we found that equidistant RPE-levels do not map directly on equidistant handgrip force. In line with previous research, equidistance was found at lower effort levels (i.e., same size differences in produced force when comparing 20–40% with 40–60%) but not at high effort levels (i.e., a larger difference in produced force in the 60–80% comparison when contrasted with the 20–40% and 40–60% comparison) [29, 31]. The difference in produced force between a perceived effort of 60% and 80% was twice as much as the difference between the lower RPE intervals (20–40, 40–60), indicating that different factors contribute to the perception of effort at different levels of the RPE scale. This finding is consistent with other research that has found effort perception and produced force to be not aligned in the same way across all effort levels. Interestingly unplanned exploratory post-hoc analyses revealed that at higher effort levels force production was more variable. Descriptively, the pattern mirrored the pattern found with respect to the mean forces, indicating a markedly increased variance at the 80% level. Taken together, these results point towards marked differences in how subjective effort aligns to objective performance at the upper end of the RPE scale: alignment seems to be rather straightforward and consistent at low to moderate subjective effort levels, and this mapping becomes much more variable at high effort levels. From an applied perspective, this implies that prescribing intensity at low to moderate RPE levels should lead to relatively consistent force production, whereas an exerciser might produce vastly different forces at high subjective effort levels.

Second, consistent with other research, we found that producing force at constant RPE-levels led to a decline in force production over time [32]. Interestingly, this effect depended on the effort participants were asked to produce. When asked to produce a force that reflected subjective efforts of 20% or 40%, the produced force declined over time, while no such decline was noted for subjective efforts of 60% or 80%. This indicates that the degree of subjective and objective alignment is sensitive to effort duration and intensity.

Third, we found that alignment was relatively robust against prospective effort requirements in another domain: The expectation of a subsequent mental effort task did not robustly alter the alignment between subjective effort and produced force. Thus, being aware that one would have to employ mental effort after a bout of physical effort did not generally change the produced force. Interestingly, although no main effect for subsequent effort requirements in the mental domain was found, we found a small but significant three-way interaction. This interaction revealed that prospective mental effort effects on the alignment between subjective effort and produced force were altered as a function of the required effort level and the

duration of the experiment. Intriguingly, compared with the no-effort condition, participants produced more force at subjective effort levels of 40%, 60%, and 80% until halfway through the experiment in the high and low mental effort conditions. This indicates that prospective mental effort might alter the alignment between subjective physical effort and objective performance under some but not all conditions. In the following, we discuss our studies' three main findings.

## Implications

**Alignment of perceived effort and objective behavioral output.**    Here, we replicate that perceived effort and objective performance do not straightforwardly map onto one another. Consistent with previous research, the alignment differed at different RPE-levels [28–31]. This means that perceived effort and objective performance are more aligned at some RPE-levels and less aligned at others, indicating that effort costs do not increase linearly. Reward-based choice theories claim that physiological and psychological variables likely contribute to differential increases in effort costs and the attendant differences in produced force. With respect to physiological variables, research shows, for example, that muscle fiber recruitment differs across effort levels [51, 52]. One speculative interpretation of the relatively large difference in force production from 60%-80% might reflect the additional recruitment of Type 2 muscles fibers, thereby altering the perceived effort costs at higher levels. Differential muscle fiber recruitment might also affect the precision with which people can produce force according to their subjective perception of effort, as research shows that type 1 fibers allow for more graded and controlled contractions [e.g., 53, 54]. Through this lens, it is plausible that the alignment across effort levels might not be equivalent.

**Alignment and the role of time.**    This reasoning can be extended to the observation that produced force declined over time and that this decline depended on the required level of subjective effort. Through a reward-based choice lens, changes in physiological (e.g., reduced actual capacity as reflected in a lowered MVC) and psychological states (e.g., increased boredom) should affect perceived momentary control costs in a highly dynamic fashion. For example, MVC is expected to decline over repeated submaximal contractions over time [e.g., 33]. One tentative explanation for the lower forces we observed over time could be that participants dynamically adjusted their efforts according to the gradual decline in MVC. Importantly, force declines were only observed at subjective efforts of 20% and 40% but not at 60% or 80%. This suggests that a gradual reduction in MVC was not the only reason for lower force production over time and that additional factors might also affect this decline. One speculative interpretation for the force decline at lower intensities might be that easy trials felt more boring than the higher effort trials. Consistent with this, research has frequently linked the experience of boredom to underchallenge in monotonous tasks [55, 56]. In addition, recent work on physical effort showed that low effort tasks can feel more boring than high effort tasks [57]. In addition, unpublished work from out lab indicates that people feel that they have to exert additional effort due to boredom. Unfortunately, this interpretation requires future research because we did not track boredom throughout this experiment.

**Alignment as a function of added prospective mental effort costs.**    Interestingly, we found that the alignment between subjective effort and produced force was only marginally affected by the prospect of having to engage in subsequent mental effort. If participants conceptualize mental and physical effort as one general cost factor per trial, but still see the mental and physical task as two distinct tasks, then we would have expected that subsequent mental effort would prompt participants to withhold physical effort. However, in contrast with our expectations, during the first part of the experiment, participants produced more force when

they expected to perform a mental task afterwards. One potential explanation for this finding might be that the physical and mental task were perceived as one connected task, resulting in a shared effort investment in both tasks. Consequently, a potential explanation for this finding might be the concept of automatic effort mobilization [58, 59]. This concept suggests that the anticipation of (any) effort and, therefore, an "action prime" leads to increased sympathetic arousal and a higher mobilization of resources compared to when no subsequent effort was required (i.e., an "inaction prime"). However, the mobilization of effort should depend on how much effort a person feels is justified by the reward value of the task. To make sure that the participants' level of effort was not affected by rewards for good performance, they received a flat payment that was independent of their performance. Additionally, we provided no feedback on the quality of their performance in the physical task. Still, the feedback provided on mental task performance might have served as an incentive, enhancing the perceived rewards in the task configurations where a mental task is anticipated, leading to a higher effort investment. One factor that might contribute to higher force production in the trials with an anticipated mental task might be "need for cognition" [60–62]. This refers to the tendency to engage in cognitively challenging tasks and university students are assumed to score high on this tendency to value cognitive effort [63]. Thus, one might speculate that the prospect of a mental task trial increased the reward value of the task, thereby licencing a relatively higher investment of effort. However, recent work that has investigated the domain-specificty of peoples' effort perception showed that high need for cognition is not strongly related to peoples willingness to mobilize physical effort [64]. Taken together, these results suggest that participants demonstrate a high flexibility in their effort regulation, making fine-graded adjustments based on the integration of various sensations. However, it is important to note that the magnitude of these effects is small, and our results do not indicate that prospective mental effort factors are integrated into the computation of current physical effort costs in a straightforward way. Further research is needed to assess whether this is because people treat mental and physical effort as separate cost factors or due to methodological constraints in our research design. In future studies, it will also be interesting to assess how individual differences in peoples perception physical and mental effort alter the alignment between subjective effort and objective performance across different effort domains.

## Limitations and future study considerations

Even though the present study provided valuable insights into the aligment between subjective effort and physical performance, as well as effort regulation across domains, some limitations should be acknowledged. Concerning our research design, one potential drawback is that from a participant's point of view, the trials felt not fully separated. Thus, performance in the mental task on one trial might have altered physical performance on a subsequent trial. While this cannot be ruled out, it is important to note that we undertook measures to minimize the potential for such carry-over effects. For example, after the 4.5 seconds of mental effort, 7.5 seconds elapsed until the physical effort in the subsequent trial, and the instruction made it clear to participants when a new trial started. Nevertheless, it might be worthwhile to space out the trials even more in future research. Another aspect is that we cannot rule out that the lack of incentive may lead to an overestimation of certain subjective effort perceptions. An experiment testing this assumption could use a monetary reward for participants' precision in reaching 20/40/60/80% of their MVC without feedback and see whether the results are the same.

We assessed four out of a possible ten 10%-steps in RPE target levels. Using four RPE-levels, we can compare the alignment across a wide effort range while still making it feasible to investigate the temporal dynamics at these levels and the effects of subsequent mental effort.

Increasing the number of target levels would make the experiment even longer, thereby placing additional demands on participants. Nonetheless, future research should examine the alignment using all 10 levels to determine whether this entails any changes in the shape of alignment. For example, it is conceivable that parabolic or hyperbolic functions would better describe the alignment between RPE and produced force, or that the shape of alignment shifts as a function of time (e.g., a linear alignment in the beginning an a hyperbolic fit towards the end).

Current research argues that force accuracy between intended and actual force can be increased by visual feedback [65, 66]. However, providing feedback during force production would have changed our research question substantially: Our study focused on the actual alignment of perceived effort and power output. With performance feedback, we would have tapped more into participants' ability to correctly monitor their produced force relative to an external target and correct their mistakes accordingly, thereby potentially decoupling their behavior from their internal sensations. In addition, when examining the extent to which subjective effort translates into objective performance, it is hard to think of performance feedback in terms of good or bad performance. Consider this scenario: a participant generates a force equivalent to 30% of his MVC, despite explicit instructions to exert 60% MVC. In the context of our investigation, this is not necessarily wrong. Even though the designated target force was not met, this measure might still be valid if the person produced a force that, in their momentary perception, aligned with a subjective effort of 60%.

We focused on manipulating perceived costs by manipulating RPE-levels and anticipated mental tasks. During the task, we did not ask for perceived changes in boredom or value–variables that can potentially contribute to the perceived value of the EVC, interacting with changes in the produced force. Therefore, one should be cautious in attributing our findings solely to changes in perceived control costs because rewards can also change during experimental duration even though it was not explicitly manipulated [67]. This is particularly relevant as recent work suggests that participants generally experience boredom during study participation, leading them to alter their behavior [68]. Future research should investigate whether these rewards lead to less pronounced decreases in force generation over time. Additionally, examining whether obtained rewards can potentially counteract the perceived control costs would be worthwhile. Moreover, contrary to the popular presumption that exerting effort carries inherent costs [9], the effort paradox argues that investing effort itself can add value and be experienced as rewarding [67].

Finally, we did not measure MVC levels throughout the experiment to assess the error rate better, given that momentary MVC may change over time. It may be that subjects can make more subtle adjustments with lower RPEs than higher RPEs since they produced lower forces over time, which aligns with the notion that fatigue played a role and that their MVCs were reduced. Importantly, in this study, we were interested in how subjective effort differences robustly translate into objective differences in produced force. As the to-be-produced forces were prompted in randomized order, the relative differences between the four levels are unlikely to be driven by the temporal changes in MVC.

## Conclusion

Here, we show that the mapping of subjective effort onto objective force production is not fixed across effort levels, timescales, and domains. This suggests that RPE is sensitive to external (e.g., future mental demands, time demands) and internal factors (e.g., fatigue), whose integration could lead to dynamic adjustments in the mapping between subjective effort and objective outputs. Understanding the degree and shape of how subjective experience maps

onto objective performance and the moderators of this alignment is crucial from a theoretical and applied perspective. For the former, deeper insights into the shape and consistency of how subjective effort and objective performance are coupled can tell us more about how people integrate effort costs across levels, time periods, and domains. Concerning the latter, exercise prescription based on subjective ratings of effort might become even more effective if the alignment is better understood.

## Author Contributions

**Conceptualization:** Sebastian Schindler, Wanja Wolff.

**Formal analysis:** Katja Rewitz, Wanja Wolff.

**Investigation:** Katja Rewitz.

**Methodology:** Katja Rewitz, Sebastian Schindler, Wanja Wolff.

**Project administration:** Sebastian Schindler, Wanja Wolff.

**Resources:** Sebastian Schindler, Wanja Wolff.

**Software:** Wanja Wolff.

**Supervision:** Wanja Wolff.

**Validation:** Sebastian Schindler, Wanja Wolff.

**Visualization:** Katja Rewitz, Wanja Wolff.

**Writing – original draft:** Katja Rewitz.

**Writing – review & editing:** Sebastian Schindler, Wanja Wolff.

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
