## [Decision Letter · Decision Letter 0]

30 Apr 2024

PONE-D-23-39694Examining the Alignment between Subjective Effort and Objective Force ProductionPLOS ONE

Dear Dr. Schindler,

Thank you for submitting your manuscript to PLOS ONE. After careful consideration, we feel that it has merit but does not fully meet PLOS ONE’s publication criteria as it currently stands. Therefore, we invite you to submit a revised version of the manuscript that addresses the points raised during the review process.

Both reviewers have highlighted issues with methodological explanations and the novelty of the findings. They recommend more precise data presentation, improved framing of hypotheses, and a deeper discussion of results to enhance the manuscript's quality. One reviewer questions the relevance of the Expected Value of Control (EVC) model to this study, while another believes the discussion on it is excessive. Therefore, I recommend that the authors either clarify the relevance of the EVC model to this study or reduce its emphasis to an amount that is appropriate for this research.

We look forward to receiving your revised manuscript.

Kind regards,

Rei Akaishi

Academic Editor

PLOS ONE

Additional Editor Comments:

Both reviewers have highlighted issues with methodological explanations and the novelty of the findings. They recommend more precise data presentation, improved framing of hypotheses, and a deeper discussion of results to enhance the manuscript's quality. One reviewer questions the relevance of the Expected Value of Control (EVC) model to this study, while another believes the discussion on it is excessive. Therefore, I recommend that the authors either clarify the relevance of the EVC model to this study or reduce its emphasis to an amount that is appropriate for this research.

Reviewers' comments:

Reviewer's Responses to Questions

**Comments to the Author**

1. Is the manuscript technically sound, and do the data support the conclusions?

Reviewer #1: Yes

Reviewer #2: Partly

2. Has the statistical analysis been performed appropriately and rigorously? 

Reviewer #1: Yes

Reviewer #2: Yes

3. Have the authors made all data underlying the findings in their manuscript fully available?

Reviewer #1: Yes

Reviewer #2: Yes

4. Is the manuscript presented in an intelligible fashion and written in standard English?

Reviewer #1: Yes

Reviewer #2: Yes

5. Review Comments to the Author

Reviewer #1: The study is interesting and well-conducted, and certainly worthy of publication. Yet, I have identified several areas that require further consideration and clarification:

1. In the abstract, while well-written, you present the results as somewhat binary. It would be beneficial to elaborate on the magnitudes of these inconsistencies and their meaningfulness or triviality.

2. Under the introduction, defining "mental exercise" briefly would enhance clarity, and it's unclear why it's italicized. The extensive review of cognitive control theories might not align with the study's focus, as it revolves around perceived effort and behavioral output alignment.

3. The rationale for providing feedback on the mental task needs to be clarified.

4. The setup of the experiment is a little confusing. Were the X% randomized? Did, for example, all reps with 20% come in one block? Please clarify this point.

5. In the statistical analysis section, it's unclear what the outcome was for the one-way ANOVA. Clarifying whether it was the differences in forces between those produced and calculated based on the MVC is necessary.

6. In the results section, you should acknowledge the impact of the normalization procedure on the low values (i.e., mean normalized to peaks).

7. Under “Alignment of subjective effort level and objective force,” Maybe add the actual values in the text rather than just on the graph. It seems as if the error (i.e., RPE-based forces – calculated % based on MVC) is around 5% for the 2-4 and 4-6 conditions and increases to roughly 10% for 6-8. There is little discussion over these values, yet I think they are very important.

8. The discussion section could benefit from shortening the second paragraph and considering another term for "equidistant." Additionally, discussing the magnitude of the error and its importance would enhance the interpretation of the findings.

9. Discussing the (mis)alignment should also consider directionality. If subjects produced more rather than less force, then I fail to see how opportunity costs or pain, for example, are a good explanation.

10. A limitation that should be mentioned is the lack of post-protocol MVC, or better yet, MVCs performed throughout the protocol to better assess the error rate, given that MVCs were expected to change. Based on your results, it may be that subjects are able to make more subtle adjustments with lower RPEs than higher RPEs, since they produced lower forces over time, which aligns with the sensible notion that fatigue played a role and that their MVCs were reduced.

Reviewer #2: The review has been added as an attachment because it exceeds 20,000 characters. Main message is copied below but please refer to the attached document for more details.

The manuscript has two main focuses. One is to study the mismatch between objective and subjective force exertion and how this mismatch can be altered by fatigue/time-on-task. The second focus is to build upon the idea that physical effort depends on the expected value of control (EVC) model developed by Shenhav, Botvinick and Cohen in 2013, as has already been suggested in previous publications, and to test whether effort is allocated differently when one anticipates to exert only physical effort versus when one anticipates to exert both physical and mental effort, with the idea that both could share a limited and quickly depletable resource. The three specific claims are that 1) subjective ratings of perceived exertion (RPE) are not matching objective force intensity linearly in a handgrip squeezing task, especially for high (60-80%) force levels, 2) time induced a reduction in the objective force exerted over time for constant RPE levels, and 3) that there is an interaction between time-on-task, and the anticipation of physical effort alone vs the anticipation of physical and mental effort.

Overall, I found the article well-written and easy to follow and read. I think that some of the results are quite interesting, either because they replicate some results (claims 1 and 2) with some new results specific to their study (claim 2) or because they discovered something new that could reflect interesting processes about how the brain proceeds with effort investment (claim 3). However, I think that the current version has some major flows, especially regarding the methodological description and the way the results are presented, and I believe that a major revision could help to significantly improve the quality of the paper.

6. PLOS authors have the option to publish the peer review history of their article (what does this mean?). If published, this will include your full peer review and any attached files.

Reviewer #1: **Yes: **Israel Halperin

Reviewer #2: No

---

## [Author Response · Author response to Decision Letter 0]

27 Jun 2024

Dear Dr. Akaishi,

we would like to thank you and the two reviewers for your time and the very helpful and constructive evaluation of our manuscript. Enclosed you will find the carefully revised second version of the manuscript "Examining the Alignment between Subjective Effort and Objective Force Production”and a point-by-point response to the concerns raised and suggestions made. Responses are in bold blue font, and major insertions to the manuscript are highlighted in italics in the response letter (only visible in the attached letter, not in the online form). Additionally, insertions to the manuscript are highlighted by track changes. We also added the requested clean manuscript version. 

We are very much looking forward to your feedback.

Sincerely,

Katja Rewitz; Dynamics of Human Performance Regulation Laboratory, Insitute of Movement Science, University of Hamburg, Germany 

katja.rewitz@uni-hamburg.de

Wanja Wolff; Dynamics of Human Performance Regulation Laboratory, Insitute of Movement Science, University of Hamburg, Germany

wanja.wolff@uni-hamburg.de

Sebastian Schindler; Institute of Medical Psychology and Systems Neuroscience, Münster, Germany, Sebastian.Schindler@ukmuenster.de

Review Comments to the Author

Reviewer #1: The study is interesting and well-conducted, and certainly worthy of publication. Yet, I have identified several areas that require further consideration and clarification:

Thank you for the positive overall evaluation!

1. In the abstract, while well-written, you present the results as somewhat binary. It would be beneficial to elaborate on the magnitudes of these inconsistencies and their meaningfulness or triviality.

We agree that the prior version of the abstract was not comprehensive enough with respect to what we specifically found and who this might be meaningful for. We have now rewritten this according to your comment. 

“[…] We found that the RPE-force alignment was not the same across RPE-levels: Whereas subjective differences from 20-40% and 40-60% were met by comparable differences in produced force, a substantially larger difference was observed for the 60-80% interval. Interestingly, exploratory post-hoc analyses revealed that this was mirrored by an increase in variance at the higher effort levels. In addition, at constant RPE-levels, participants produced less force over time, and this effect was more pronounced at lower RPE target levels. Lastly, anticipating mental effort after the physical effort slightly altered the alignment as a function of the to-be-produced RPE-level and experimental duration. Taken together, our results indicate that the mapping of perceived effort on objective performance is intricate, and several factors affect the degree and shape of how RPE and performance align. Understanding the dynamic adjustment of RPE-performance alignment across different RPE levels is particularly relevant for contexts that use RPE as a tool for training load prescription.“

2. Under the introduction, defining "mental exercise" briefly would enhance clarity, and it's unclear why it's italicized. 

Thank you for pointing out that “mental exercise” as a term may not be clear in this context. Here, we use “mental exercise” as a counterpart to physical exercise. The term is more commonly found in lay than scientific literature, so we initially decided to italicize it. By “mental exercise”, we refer to any engagement in cognitively stimulating activities. Because we do not want to imply that mental exercise is a scientific term, we now clarify this by explicitly stating that mental exercise is a “lay term” and explaining what we are referring to in the text. 

“Physical exercise is associated with a reduction of health-related risk factors, higher quality of life, and reduced mortality (Chastin et al., 2019; Hummel et al., 2022; Hupin et al., 2015). Mental exercise is a frequently used lay term describing the engagement in mentally stimulating activities and is suggested to contribute to developing a "cognitive reserve". This reserve is posited to be a protective factor against cognitive decline and clinical symptoms in the early onset of neurological diseases such as Alzheimer's (Pettigrew & Soldan, 2019; Stern, 2009).”

3. The extensive review of cognitive control theories might not align with the study's focus, as it revolves around perceived effort and behavioral output alignment.

Thank you for highlighting that an extensive review of cognitive control theories might not be necessary for our study’s focus. We agree and have, therefore, shortened the review. To facilitate understanding of our assumptions and hypotheses, we stick to the EVC-theory as an example of effort allocation as a value-based decision and potential changes in effort investment associated with different effort demands or changes in perceived costs over the course of the study. However, we now clarified that most reward-based choice theories in motor and cognitive control research make predictions that are similar with respect to their key premises (that are relevant for us). Namely: Effort carries a cost that people try to minimize and only exert to the degree that it is justified by value/reward. The substantially shortened section now reads: 

“Cognitive control, also referred to as self-control, is defined as "the set of mechanisms required to pursue a goal, especially when distraction and/or strong competing responses must be overcome" (Shenhav et al., 2013, p. 217). Applying control is perceived as effortful (Cohen et al., 1990), and perceived effort is understood to index the costs of applying control (Shenhav et al., 2017). In the present study, we adapt the definition of perceived effort as “the instantaneous experience of investing resources to perform an action” (Halperin & Vigotsky, 2024, p. 8). In line with this definition we understand the perception of effort to be based on the integration of various contributing factors rather than being solely contingent on objective behavioral output (e.g., force production) (Halperin & Vigotsky, 2024). 

Motor control and cognitive control theories suggest that the execution of a control action reflects a reward-based choice that is geared towards minimizing control costs (Wolpert, 1997; Shenhav et al., 2013). To illustrate, according to one influential framework—the Expected Value of Control Theory (EVC theory)—the expected value of control is calculated for all available behavioral options by discounting the expected positive and negative payoffs that are associated with task engagement and the intrinsic cost of control. These intrinsic costs scale with task demands and task duration (Shenhav et al., 2013). Thus, harder and longer tasks are expected to be more costly than short and easy ones (Wolff et al., 2021). During task performance, environmental information (e.g., feedback, opportunity for rewards) and internal sensations (e.g., pain, motivational loss, boredom) are assumed to be monitored, updated, and integrated to initiate adjustments in effort allocation (Geana et al., 2016; Shenhav et al., 2013; Shenhav et al, 2016). Consequently, if changes in perceived control costs alter this cost-benefit analysis, performance decrements or even task termination are expected to ensue (Shenhav et al., 2013). 

Empirical evidence demonstrates that the regulation of mental and physical effort can be effectively described by theories relying on the concept of a cost-benefit analysis (Manohar et al., 2015; Wolff et al., 2020). However, recent theoretical and empirical advancements notwithstanding, the relationship between perceived effort as an index of control costs and the translation of this perception into objective behavioral output is not fully understood. Simply put, we do not know how perceived effort levels and objective output align and what factors affect this alignment. In this study, we will investigate these questions through the lens of reward-based choice theories of control.”

4. The rationale for providing feedback on the mental task needs to be clarified 

Thank you for pointing out this omission. We have now clarified this: 

“After each block, performance feedback was given regarding the accuracy of responses across all mental tasks from the preceding experimental block. Feedback was provided as a self-monitoring device so participants could make sure that they were correctly performing the different mental tasks throughout the 2-hour experiment. The feedback provided was not tied to specific trials. Hence, participants were not given any information on their performance regarding different task difficulties.”

5. The setup of the experiment is a little confusing. Were the X% randomized? Did, for example, all reps with 20% come in one block? Please clarify this point.

Thank you for asking. Conditions were randomized. Thus, not all reps with 20% were performed consecutively. We clarified this in the methods section.

“Participants were asked to produce four force levels in alignment with predetermined target levels of a modified Borg CR-10 scale (Borg, 1982). The Borg CR-10 scale is a 10-point category-ratio-scale used to measure exertion and pain (Capodaglio, 2001). The upper anchor of 10 represents the maximal possible exertion. The target levels 2, 4, 6, and 8 were selected to cover nearly the whole scale range. Participants were instructed that these levels should represent 20%, 40%, 60%, and 80% of their maximum. The target levels were presented in a random order throughout the experiment. During the whole experiment, participants performed 240 submaximal contractions.” 

6. In the statistical analysis section, it's unclear what the outcome was for the one-way ANOVA. Clarifying whether it was the differences in forces between those produced and calculated based on the MVC is necessary.

We fully agree that this was not worded clearly enough in the prior submission. We have now provided a more precise description of what was tested in the ANOVA. This has clarified our reasoning and intended outcome. 

“To investigate whether the subjective differences of the RPE scale intervals transform into the same intervals in objective force, a one-way repeated measures ANOVA was performed with the averaged differences in produced force (expressed in %MVC) between target RPE levels as the dependent variable. Thus, we compared whether the relative difference in produced force for the three equal-sized subjective intervals of effort (i.e., 20-40%, 40-60%, and 60-80%), each reflecting a 20% increase in subjective effort, translate into equivalently sized intervals in produced force.”

7. In the results section, you should acknowledge the impact of the normalization procedure on the low values (i.e., mean normalized to peaks).

Good point! We have updated the manuscript accordingly: 

“In physical effort level 2, force production differed M = 11.19% (SD = 6.66%), in level 4 M = 24.93% (SD = 8.59%), in level 6 M = 39.31% (SD = 10.53%) and in level 8 M = 46.86% (SD = 14.18%) from target level MVC. This underproduction relative to peak MVC is likely due to the fact that target level MVC was calculated as a proportion of the achieved peak force, whereas force production was averaged across 2 seconds.”

8. Under “Alignment of subjective effort level and objective force,” Maybe add the actual values in the text rather than just on the graph. It seems as if the error (i.e., RPE-based forces – calculated % based on MVC) is around 5% for the 2-4 and 4-6 conditions and increases to roughly 10% for 6-8. There is little discussion over these values, yet I think they are very important.

Thank you. We added the actual values in the text to make the non-equivalent differences in produced force between RPE target intervals easier to understand. 

“[…]A significant difference was found when comparing the averaged deviations between the RPE-level intervals (F(1.36, 57.14) = 63.85, p < .001, η_g^2= 0.34). The deviations represent the differences in the averaged produced force (in % MVC) between target level 20% compared to 40% (M = 6.26%, SD= 3.18%), 40% compared to 60% (M = 5.62%, SD = 3.74%) and 60% compared to 80% (M = 12.45%, SD = 5.58%). Bonferroni corrected post-hoc analyses revealed a significant difference in produced force relative to MVC between the target levels 20% and 40% compared to 60% and 80% (t(42) = -7.43, p < .001) and between the target levels of 40% and 60% in comparison with 60% and 80% (t(42) = -9.92, p < .001). The deviation between the 20% and 40% target levels did not differ from the deviation between the target levels of 40% and 60% (p = .39).[…]”

9. The discussion section could benefit from shortening the second paragraph and considering another term for "equidistant." Additionally, discussing the magnitude of the error and its importance would enhance the interpretation of the findings.

We agree that expanding on the increased variance at higher effort levels is important here! We have now added an analysis of the error variance to the paper (clearly marked as an unplanned exploratory post-hoc analysis) and we now discuss this in the second paragraph of the discussion. Particularly, we discuss what this might mean for applied settings. Our reading of this finding is that prescribing intensity at low to moderate RPE levels should lead to relatively consistent force production (due to lower variance), whereas an exerciser might produce vastly different forces at high subjective effort levels (due to higher variance). 

(Regarding the term ‘equidistant’: we have considered another term for this but after careful deliberation and also looking for other terms in the literature, we would prefer to keep using the term. This is because we feel it accurately describes what we refer to and because it is used in other studies too. However, we are clearly not ‘married’ to it, so if you insist on it being not the adequate term here, then we are happy to change it in another round of revisions) 

Thanks again for pointing us towards the relevance of looking into the errors more: 

“[…]Error bars in Figure 2 indicate that the variances in produced forces are not the same for each effort level. Consequently, we conducted an unplanned exploratory post-hoc analysis to assess whether variances differed significantly between levels. Consistent with the descriptive trends in Figure 2, a repeated measures ANOVA and post hoc comparisions revealed that variance increased at higher effort levels F(1.32, 55.26) = 67.96, p < .001), with a significantly higher variance at target level 80% compared to the lower effort levels (p < .001). For the full analysis, please see OSF | alignment subjective effort_objective force production.”

“First, we found that equidistant RPE-levels do not map directly on equidistant handgrip force. In line with previous research, equidistance was found at lower effort levels (i.e., same size differences in objective force when comparing 20-40% with 40-60%) but not at high effort levels (i.e., a larger difference in objective force in the 60-80% comparison when contrasted with the 20-40% and 40-60% comparison)(Cochran et al., 2007; Pincivero, 2011). The difference in produced force between a perceived effort of 60% and 80% was twice as much as the difference between the lower RPE intervals (20-40, 40-60), indicating that different factors contribute to the perception of effort at different levels of the RPE scale. This finding is consistent with other research that has found effort perception and produced force to be not aligned in the same way across all effort levels. Interestingly unplanned exploratory post-hoc analyses revealed that at higher effort levels force production was more variable. Descriptively, the pattern mirrored the pattern found with respect to the mean forces, indicating a markedly increased variance at the 80% level. Taken together, these results point towards marked differences in how subjective effort aligns to objective performance at the upper end of the RPE scale: alignment seems to be rather straightforward and consistent at low to moderate subjective effort levels, and this mapping becomes much more variable at high effort levels. From an applied perspective, this implies that prescribing

---

## [Decision Letter · Decision Letter 1]

16 Jul 2024

Examining the Alignment between Subjective Effort and Objective Force Production

PONE-D-23-39694R1

Dear Dr. Schindler,

We’re pleased to inform you that your manuscript has been judged scientifically suitable for publication and will be formally accepted for publication once it meets all outstanding technical requirements.

Kind regards,

Rei Akaishi

Academic Editor

PLOS ONE

Additional Editor Comments (optional):

The paper is basically accepted. Reviewer 2 made an important comment. Please follow the major comment and minor suggestions of the reviewer 2.

Reviewers' comments:

Reviewer's Responses to Questions

**Comments to the Author**

1. If the authors have adequately addressed your comments raised in a previous round of review and you feel that this manuscript is now acceptable for publication, you may indicate that here to bypass the “Comments to the Author” section, enter your conflict of interest statement in the “Confidential to Editor” section, and submit your "Accept" recommendation.

Reviewer #1: All comments have been addressed

Reviewer #2: (No Response)

2. Is the manuscript technically sound, and do the data support the conclusions?

Reviewer #1: Yes

Reviewer #2: Yes

3. Has the statistical analysis been performed appropriately and rigorously? 

Reviewer #1: Yes

Reviewer #2: Yes

4. Have the authors made all data underlying the findings in their manuscript fully available?

Reviewer #1: (No Response)

Reviewer #2: Yes

5. Is the manuscript presented in an intelligible fashion and written in standard English?

Reviewer #1: (No Response)

Reviewer #2: Yes

6. Review Comments to the Author

Reviewer #1: You have done an excellent job with the revisions. I have nothing else to comment about. Congratulation on this important article.

Israel Halperin

Reviewer #2: The authors have addressed most of my questions and my requests for additional information, with very helpful edits to the text and figures that clarify aspects of the methods and findings. I think that the manuscript is now much better, more accessible to the reader and will form a valuable addition to the literature on the topic. I have one main last comment that I think deserves deeper consideration and a few more minor comments that I leave open to the editor and authors to consider or to ignore.

Based on the way the task was instructed and incentivized, the only reward for the participants which pushes them to produce some force is to please the experimenter and to feel that they are “doing the task well”. I was therefore wondering whether, as an EVC model would state, they are not trying to minimize the force production, while still displaying some variation in the force produced to please the experimenter based on their expectation of what the experimenter wants to observe. This could explain the global difference between force produced and the RPE. As the authors replied to my 11th comment “we cannot rule out that the lack of incentive may lead to an overestimation of certain subjective effort perceptions”. One way to target this would be to perform an experiment where participants would be rewarded for their precision in reaching 20/40/60/80% of their MVC with no feedback and see whether the results are the same. However, I do acknowledge that any difference in the results could also stem from the fact that such an experiment would reward performance instead of the feeling of effort which is intrinsically subjective, as discussed in our previous exchanges, but if the two were results were consistent that would form a solid ground about the results obtained. In any case, I would include the sentence “we cannot rule out that the lack of incentive may lead to an overestimation of certain subjective effort perceptions” somewhere in the discussion to acknowledge that the global shift between force produced and subjective RPE could also rely on the task incentives.

Minor:

• Regarding the way force was measured (2s average):

- I thank the authors for explaining the instructions that were not so clear to me. I thought that participants had to produce a peak force within a 4s period, and not to maintain that force during this period. To avoid confusion for other readers, I would strongly recommend to precise this information in the Methods.

- I also thank the authors for providing the data re-analyzed using the peak force instead of the averaged force which rules out any potential confound in the results that could be due to the way force was extracted. I think that this figure could be included as a supplementary result in case there are other readers who have the same concern, but I also acknowledge that, given the instructions, it may not be as relevant as I initially thought.

- Based on the authors’ explanations, I completely understand now that an average was taken to extract the produced force, and that this average ignores the beginning of the contraction period because it takes time for the subjects to reach the target force level, but I struggle to see why the last samples were also ignored. In principle, I would have thought that the participants should not release their squeeze until the end of the trial but maybe I missed something in the instructions? Or maybe the authors just observed that there was more variability towards the end of the trial and preferred to discard them? In any case, a short clarification would be helpful to help the reader to know why the last part of the trial has also been removed.

• Figure 2: I really appreciate the change which I think makes the result much clearer. I have two small suggestions to improve that figure even further, but the authors are welcome to ignore them if they don’t think that they are valid:

- Write the % of deviation between each RPE level on the curve (which would highlight the result that the authors want to emphasize regarding a 20% change in RPE not mapping to a 20% change in force produced linearly)

- Add a theoretical curve reflecting the “ideal” performance with a perfect matching between target RPE levels and produced force to highlight how much the produced force differs from the RPE.

• Sex differences: I still think that it would be important to at least look at the data after splitting for sex just to ascertain that the effects go in the same direction and maybe add it in Supplementary Material, but I leave that decision to the Editor and the authors.

7. PLOS authors have the option to publish the peer review history of their article (what does this mean?). If published, this will include your full peer review and any attached files.

Reviewer #1: **Yes: **Israel Halperin

Reviewer #2: No

---

## [Editor Report · Acceptance letter]

31 Jul 2024

PONE-D-23-39694R1 

PLOS ONE

Dear Dr. Schindler, 

I'm pleased to inform you that your manuscript has been deemed suitable for publication in PLOS ONE. Congratulations! Your manuscript is now being handed over to our production team.

Kind regards, 

on behalf of

Dr. Rei Akaishi 

Academic Editor

PLOS ONE